# WINNING PRIVATELY: THE DIFFERENTIALLY PRIVATE LOTTERY TICKET MECHANISM

## ABSTRACT

We propose the differentially private lottery ticket mechanism (DPLTM). An end-to-end differentially private training paradigm based on the lottery ticket hypothesis. Using "high-quality winners", selected via our custom score function, DPLTM significantly outperforms state-of-the-art. We show that DPLTM converges faster, allowing for early stopping with reduced privacy budget consumption. We further show that the tickets from DPLTM are transferable across datasets, domains, and architectures. Our extensive evaluation on several public datasets provides evidence to our claims.

## 1 INTRODUCTION

Learning while preserving the privacy of the contributing users is a priority for neural networks trained on sensitive data. Especially, when it is known that neural networks tend to "remember" training data instances (such as a patient's healthcare information) (Carlini et al., 2018; Fredrikson et al., 2015; Song et al., 2017; Wu et al., 2016). Differential privacy (Dwork et al., 2006) has become the *de facto* standard for protecting an individual's privacy in machine learning. Differentially private training of neural networks ensures that the model does not unduly disclose any sensitive information. The most often used approach to achieve this goal is the method of gradient perturbation, where we add controlled noise to the gradients during the training phase. Differentially Private Stochastic Gradient Descent (DPSGD) (Abadi et al., 2016) is the current state-of-the-art, used extensively for training privacy-preserving neural networks. DPSGD, however, falls short on the utility front, the main reason for which we discuss below.

DPSGD, for a given minibatch, first computes the per-observation gradient, $g(x_i)$, and then clips $g(x_i)$ in $l_2$ norm, $g(x_i)/\max(1,\|g(x_i)\|_2/C)$ (Line 6 in Abadi et al. (2016), Algorithm 1). We can see that the norm will be large (proportional to the number of model parameters), especially for a multi-layer neural network, leading to a large "clipping impact" on the gradient, resulting in "smaller" clipped-gradient magnitude, easily overwhelmed by noise, required for preserving differential privacy. This leads to diminished utility, especially, for scenarios where we require tight privacy (large noise).

An obvious solution is to minimize the number of model parameters while maximizing the model's utility. This, however, is non-trivial for differentially private neural networks, especially when we need to balance privacy and utility. Recently, it has been shown that there exist smaller sub-networks within large neural networks, which when trained in isolation provide similar utility as the large networks (Frankle & Carbin, 2018; Frankle et al., 2019). The phenomenon, known as the lottery ticket hypothesis is an encouraging step towards finding small, high-utility architectures. But, directly using the lottery ticket hypothesis with differential privacy is non-trivial as we need to ensure the complete process (from ticket selection to training the winning ticket) is end-to-end differentially private.

As a potential solution to improve the privacy-utility bottleneck in differentially private neural networks, we propose the Differentially Private Lottery Ticket Mechanism (DPLTM). To ensure differential privacy in DPLTM, we use a three-tiered approach. In phase 1, we create the lottery tickets following the principles of the lottery ticket mechanism. In phase 2, we select the winning ticket with differential privacy, making sure that we pick the ticket with a small number of parameters and with high utility via our custom score function (details in Section 3.3.1). After differentially private selection of the winning ticket, our phase 3 trains the winning architecture with differential privacy. In summary, our main contributions in this work are as follows:

1. We propose DPLTM, the differentially private lottery ticket mechanism. An end-to-end differentially private extension of the lottery ticket hypothesis. We design a custom score function to ensure the differentially private selection of "best" winners, which aids DPLTM to significantly improve the privacy-utility trade-off for differentially private neural networks.

2. Due to the reduced noise impact in DPLTM, we show that DPLTM converges at a faster rate compared to DPSGD, leading to consumption of smaller privacy budget and better utility if early stopping is desired.

3. We show that the winning tickets in DPLTM are "transferable" across datasets, domains, and model architectures. Leading to the possibility of ticket generation and selection on any public dataset, resulting in lower privacy cost.

4. Using five real-life datasets, we show that DPLTM significantly outperforms DPSGD on all datasets, and for all privacy budgets.

## 2 PRELIMINARIES

We use this section to introduce differential privacy and the lottery ticket hypothesis.

### 2.1 DIFFERENTIAL PRIVACY

Differential privacy (Dwork et al., 2006) provides us with formal and provable privacy guarantees, with the intuition that a randomized algorithm behaves similarly on "similar" input datasets, formally

**Definition 1.** (Differential privacy (Dwork et al., 2006)) *A randomized mechanism $\mathcal{M} : D^n \to \mathbb{R}^d$ preserves $(\epsilon, \delta)$-differentially privacy if for any pair of neighbouring databases $(x, y \in D^n)$ such that $d(x, y) = 1$, and for all sets $\mathcal{S}$ of possible outputs:*

$$Pr[\mathcal{M}(x) \in \mathcal{S}] \le e^\epsilon Pr[\mathcal{M}(y) \in \mathcal{S}] + \delta$$

Intuitively, Definition 1 states that for any pair of two neighboring datasets, $x, y$, differing on any one row, a randomized mechanism $\mathcal{M}$'s outcome does not change by more than a multiplicative factor of $e^\epsilon$. Moreover, the guarantee fails with probability no larger than $\delta$. If $\delta = 0$, we have pure-$\epsilon$ differential privacy.

The Exponential Mechanism (EM) (McSherry & Talwar, 2007) is a well-known tool for providing differential privacy. Defined by a range $\mathcal{R}$, privacy parameter $\epsilon$, and a score function $u : \mathcal{X}^N \times \mathcal{R} \to \mathbb{R}$ that maps a dataset to the utility scores, given a dataset $D \in \mathcal{X}^N$, the EM defines a probability distribution over $\mathcal{R}$ according to the utility score. In other words, the EM is more likely to output some $r \in \mathcal{R}$ with higher utility scores, formally

**Definition 2.** (Exponential Mechanism (McSherry & Talwar, 2007)) *The exponential mechanism $\mathcal{M}(D, u, \mathcal{R})$ selects and outputs an element $r \in \mathcal{R}$ with probability proportional to*

$$\exp\left(\frac{\epsilon u(D, r)}{2\Delta u}\right)$$

*where $\Delta u$ is the sensitivity ($\Delta u = \max_{r \in \mathcal{R}} \max_{X, X':||X-X'||_1 \le 1} |u(X, r) - u(X', r)|$) of the score function.*

In terms of privacy guarantees, the exponential Mechanism provides $\epsilon$- differential privacy (McSherry & Talwar, 2007)

### 2.2 LOTTERY TICKET HYPOTHESIS

The Lottery Ticket Hypothesis was proposed by Frankle & Carbin (2018), where interestingly, it was shown that randomly initialized neural networks contain small subnetworks, which when trained in isolation, can provide similar utility as the full network. To get the subnetworks, we train a network for $\mathcal{I}$ iterations, prune $p\%$ of its weights (of smallest magnitude), and reset the weights of the pruned network to original initialization, to be trained again. This process ensures that for $n$ rounds, each round prunes $p^{1/n}\%$ of the weights. Such pruned, small subnetworks, with high utility are known as "winning tickets" from the lottery mechanism.

# 3 DIFFERENTIALLY PRIVATE LOTTERY TICKET MECHANISM

## 3.1 OVERVIEW

We start by providing an overview of DPLTM. Using the input dataset, $X$, we generate and store multiple "lottery-tickets", each with a varying number of model parameters. Then we use the exponential mechanism (McSherry & Talwar, 2007) to pick a winning ticket using a custom utility function (details follow), that provides us with the desired balance between the number of model parameters and the model utility. After selecting the winner, we train the winning architecture with differential privacy. Our total privacy cost, hence, is composed of two separate parts, selecting the winning ticket and training the winning ticket. We present the complete process succinctly as Algorithm 1 followed by a walk-through.

---

**Algorithm 1** Differentially Private Lottery Ticket Mechanism (DPLTM)

---

**Require:** Dataset: $X$, Total privacy budget: $(\epsilon, \delta)$, Pruning percent: $p$, Number of tickets: $T$, Ticket training iterations: $\mathcal{I}$, final model training iterations: $I$, Neural Network: $f$, Initial model parameters: $\theta_0$, Initial mask: $m$, Privacy budget for ticket selection: $\epsilon_1$, Privacy budget for ticket training: $\epsilon_2, \delta$, Constant for score function: $\nu$, Minibatch size: $L$, Clipping factor: $C$, Learning rate: $\eta$

---

**Phase 1 - Generating Lottery Tickets**

---

1: **procedure** LTH($f$,$m$, $\theta_0$)
2:     Randomly initialize $f(X, m \odot \theta_0); m = 1^{|\theta_0|}$                 ▷ $m$ is the initial mask
3:     **for** $i \in T$ **do**
4:         Train $f(X, m \odot \theta_0)$ for $\mathcal{I}$ iterations, to get $f(X, m \odot \theta_{\mathcal{I}})$
5:         Prune $p\%$ of parameters from $\theta_{\mathcal{I}}$, creating a new mask $m'$    ▷ $m'[j] = 0$, where $\theta_{\mathcal{I}}$ is pruned
6:         Reset the remaining parameters to their values in $\theta_0$
7:         $f(X, m' \odot \theta_0)$ is the lottery ticket
8:         Store the mask $m'$, initial parameters $\theta_0$
9:         Store the proportion of remaining model parameters, $c$, in the lottery ticket
10:        Store the performance of the lottery ticket, $a$, on the test set
11:        Let $m = m'$
12:     **end for**
13:     **return** $\mathcal{C}, \mathcal{A}, M, \theta_0$    ▷ The collection of parameter proportions, model utilities, masks, and initial parameters, $c_i \in \mathcal{C}, a_i \in \mathcal{A}, m'_i \in M; i \in [1, T]$
14: **end procedure**

---

**Phase 2 - Selecting a Winning Ticket**

---

15: **procedure** DPWT($\mathcal{C}$,$\mathcal{A}$, $\epsilon_1$)
16:     Calculate score $\mathcal{S}(\mathcal{C}, \mathcal{A}) = \mathcal{A}(1 - \nu\mathcal{C})$
17:     Select a winning ticket, $\mathcal{T}$, with probability, $P = \dfrac{\exp(\frac{\epsilon_1 \mathcal{S}}{2\Delta})}{\sum_T \exp(\frac{\epsilon_1 \mathcal{S}}{2\Delta})}$
18:     **return** $\mathcal{T}$                             ▷ The winning ticket
19: **end procedure**

---

**Phase 3 - Training the Winning Ticket**

---

20: **procedure** DPTWT($\mathcal{T}$)
21:     Initialize the network,$f$, with mask and initial values from the winning ticket
22:     **for** $k \in I$ **do**
23:         Take a minibatch with sampling probability $L/N$        ▷ $N$ is the total dataset size
24:         For each $x_i \in L$, compute gradient $g_k(x_i) = \nabla_{\theta_k} \mathcal{L}(\theta_k, x_i)$
25:         $\hat{g}_k = \frac{1}{L}(\sum_i g_k(x_i)/\max(1, \frac{||g_k(x_i)||_2}{C}) + \mathcal{N}(0, \sigma^2 C^2 I))$
26:         $\theta_{k+1} \to \theta_k - \eta_k \hat{g}_k$
27:     **end for**
28: **end procedure**

---

## 3.2 DPLTM WALKTHROUGH

Next we provide phase by phase walkthrough of DPLTM.

*Phase 1 (Generating lottery tickets)*: We start with generating lottery tickets required for DLPTM. At this stage, we are not yet concerned about privacy. So we use the non-private lottery ticket mechanism with our input dataset $X$, and generate $T$ number of lottery tickets ($t_i; i \in [1, \cdots, T]$). Specifically, we use iterative pruning version of the lottery ticket mechanism, where using the pruning parameter, $p$, at each ticket iteration, we remove $p\%$ of model parameters with the smallest magnitude. This results in $T$ tickets with a successively smaller number of parameters. Results from each ticket (accuracy on the test set, $\mathcal{A}_i, i \in [1, T]$) are stored along with the fraction of parameters ($\mathcal{C}_i, i \in [1, T]$) remaining in the model. For further use, we also store the mask $m_i'$ from each ticket and the randomly initialized initial parameters $\theta_0$.

*Phase 2 (Selecting a winning ticket with differentially privacy)*: After generating candidate tickets, we need to select a winning ticket that can be subsequently trained with differential privacy. But, picking a winner is non-trivial for two reasons, first as the tickets are generated using sensitive data, we cannot directly pick a winner, that is, we need differential privacy for selecting the winning ticket, second, we need to pick a winner such that the winner has an adequate balance between the number of model parameters (smaller the better) and the model performance (higher the better). As a solution, we use the Exponential Mechanism (EM) (McSherry & Talwar, 2007) to pick our winner with differential privacy. And to balance the number of model parameters and the utility, we define our custom score function using the combination of the accuracy achieved by the ticket on the test set and the proportion of parameters left in the network (more details in Sections 3.3.1 and 3.4).

*Phase 3 (Training the winning ticket with differential privacy)*: After we select our winning ticket with differential privacy in phase 2, now we need to train our "winner" architecture so the final model is differentially private. We do so by using the method similar to the differentially private stochastic gradient descent (DPSGD) (Abadi et al., 2016) for the training of our winning ticket. This, in contrast to DPSGD's naive implementation, now only trains a "sub-network" with a significantly small number of parameters. And hence provides significantly better utility, reasons for which we discussed at length in the Introduction. Our extensive empirical evaluation in Section 4 provides evidence for this claim.

Next, we provide formal privacy guarantees for DPLTM. We start with introducing the EM for DPLTM with our custom score function.

## 3.3 DIFFERENTIAL PRIVACY GUARANTEES OF DPLTM

As seen in Algorithm 1, our first phase of generating candidate tickets is non-private. Differential privacy comes into play in phase 2, where we pick a winning ticket with differential privacy. Hence, we start with privacy guarantees of phase 2.

### 3.3.1 SELECTING A WINNING TICKET

For the Exponential Mechanism (EM), as discussed in preliminaries, we need to define a score/utility function that assigns a higher score to "good" outputs. For DPLTM, we define the score function as follows

$$\mathcal{S}(\mathcal{C}, \mathcal{A}) = \mathcal{A} \times (1 - (\nu \mathcal{C})) \tag{1}$$

where $\mathcal{A}$ is the classification accuracy on the test set for the given network configuration (ticket), $\mathcal{C}$ is the proportion of remaining weights in the network, and $\nu$ is a constant. We discuss some properties of the score function in detail in the following section (Section 3.4).

After defining the score function, we sample our winning ticket with probability

$$P = \frac{\exp(\frac{\epsilon_1 \mathcal{S}}{2\Delta})}{\sum_T \exp(\frac{\epsilon_1 \mathcal{S}}{2\Delta})} \tag{2}$$

where $P$ is the probability of picking a ticket, $\epsilon_1$ is the privacy budget for EM, and $\Delta$ is the sensitivity of the score/utility function. Tickets with "higher" score function have a higher probability of getting selected compared to the tickets with a lower score.

**Lemma 1.** *Sensitivity ($\Delta$) of the score function, $\mathcal{S}$, is directly proportional to $\nu$.*

*Proof.* Sketch: The classification accuracy ($\mathcal{A}$), and the proportion of remaining parameters in the model ($\mathcal{C}$) are bounded by $[0, 1]$. The change of a single record can "at-worse" impact the product of ($\mathcal{A}, \mathcal{C}$) by 1. Hence the overall sensitivity is proportional to the constant, $\nu$. Full, formal proof is provided in the Appendix for space constraints. □

**Theorem 1.** *Phase 2 (Selecting a winning ticket) is ($\epsilon_1$) - differentially private.*

*Proof.* Sketch: We get the proof using our custom score function with an instantiation of EM. It is provided in the Appendix for space constraints. □

### 3.3.2 TRAINING THE WINNING TICKET

After we select our winning ticket with differential privacy in phase 2. Our next step is to train the winning architecture in a differentially private fashion. For this step, we use the training process similar to DPSGD (Abadi et al., 2016). Specifically, after calculating the per-observation gradients for a minibatch, we clip the gradients (line 25 in Algorithm 1) by their $l_2$ norm, scaled by a constant $C$, to enforce sensitivity, and then add appropriate Gaussian noise to ensure differential privacy, formally

**Theorem 2.** *Phase 3 (Training the winning ticket) is ($\epsilon_2, \delta$) - differentially private, if we chose* $\sigma \geq c \frac{L/N \sqrt{I \log(1/\delta)}}{\epsilon_2}$

*Proof.* Proof is argued in the same way as DPSGD (Abadi et al., 2016) using $\epsilon_2$ and $I$. It is omitted here for space constraints. □

### 3.3.3 PUTTING IT ALL TOGETHER

After generating the lottery tickets, selecting the winner with differential privacy, and the differentially private training of the winning ticket, we are now ready to "put it all together" and state the overall privacy guarantees of our proposed method (DPLTM).

**Theorem 3.** *Algorithm 1 is ($\epsilon, \delta$) - differentially private, with $\epsilon > 0, \delta > 0$*

*Proof.* We have already seen that phase 2 is $\epsilon_1$-differentially private and phase 3 is $\epsilon_2, \delta$-differentially private. Using the "naive" composition (Dwork et al., 2006)[1], it is easy to see that the Algorithm 1 is $\epsilon, \delta$-differentially private, with $\epsilon = \epsilon_1 + \epsilon_2$ and $\delta = \delta$. □

### 3.4 DISCUSSION

We use this section to discuss some interesting properties of DPLTM. A first observation is the seamless integration of differential privacy with the lottery ticket mechanism in DPLTM, making it accessible for implementation, while providing significantly better utility compared to the "naive" DPSGD (Experiments in Section 4 support this claim). As per earlier discussions, the success of our method is hinged on the existence of a winning ticket with *fewer* number of model parameters and high utility. So it is vital that out of all available tickets, the probability of selecting the winner with a small number of model parameters and high accuracy is high. Our custom utility function $\mathcal{S}$ strives to achieve this goal by ensuring that the selection process does not degenerate to uniform random sampling. In particular, the utility function assigns more weight to the models with high accuracy and a low number of model parameters, which we modulate using the constant $\nu$. Large $\nu$ assigns more weight to $\mathcal{C}$ (proportion of model parameters in the ticket). For good utility with tight privacy, we advocate using large values for $\nu$. [2]

As we observe from Theorem 3, total privacy budget for our method is composed of two parts. The privacy budget from the EM phase used to select the winning ticket and the privacy budget to train the winning ticket. Hence, we need to decide on the overall privacy budget split. That is, the portion of the budget to allocate to the drawing of the winning ticket and the portion of the privacy budget for

---

[1]As we are only composing two mechanisms, advanced composition is not necessary.

[2]If most tickets are performing similarly, which is often the case, it is of best interest to pick the ticket with the smallest number of model parameters.

the training of the winning ticket. We advocate dedicating a *large* proportion of the privacy budget to the training of the winning ticket and a small portion for selecting the winning ticket. With our defined utility function, a small privacy budget suffices for selecting a "good" ticket (empirical evidence provided in Section 4). A question a reader might ask: Why can't we train the full network differentially privately when generating tickets? That is, why do we produce non-private tickets first and then train a differentially private model? The answer is simple, as differential privacy composes by iteration, training multiple networks using the methodology described in Algorithm 1 would result in a large privacy budget, and hence, noisier models, providing worse utility compared to our proposed method.

## 4 EXPERIMENTS

Here we provide empirical evidence on various datasets to support our claim that our proposed method (DPLTM) outperforms DPSGD. We begin by describing the datasets.

### 4.1 DATASETS

For our empirical evaluation, we use five publicly available datasets. Dataset details are provided in Table 1.

| Dataset | Attributes | Observations | Class |
|---|---|---|---|
| MNIST | 784 | 60000 | 10 |
| HAR | 521 | 10299 | 6 |
| Fashion-MNIST | 784 | 60000 | 10 |
| Kuzushiji-MNIST | 784 | 60000 | 10 |
| ISOLET | 617 | 7797 | 26 |

Table 1: Dataset details, Attributes is the dataset dimensionality, Observations are the number of rows, and Class is the number of classes in the classification target.

### 4.2 SETUP

For generating lottery tickets (phase 1), our implementation is based on the publicly available source code[3]. Our underlying base model is a fully connected neural network with three layers. Hidden layers use ReLU (Nair & Hinton, 2010) as the activation function. Learning rate is kept fixed at 0.1 and minibatch size is kept fixed at 400. We set the pruning percent, $p$, at 30% for the first two layers and 20% for the last layer. Which means that for each subsequent ticket, the model will prune 30% of the weights compared to the previous ticket for the first two layers and 20% of the weights for the final layer. To generate lottery tickets, the mechanism is run for 5000 iterations ($\mathcal{I}$) for each ticket. Differentially private training of the winning ticket is run for 50 epochs.

DPSGD's implementation is based on the publicly available source code[4]. Clipping norm for DPSGD and DPLTM is set at a constant of 1 for all experiments. All the rest of the hyperparameters, including the underlying model architecture, are the same for DPSGD as in DPLTM to ensure a fair comparison.

If the input dataset does not have a predefined train/test partition, we use 80/20 split, with 20% of the dataset used as the test set. All models are run for 10 iterations and we report the average results. For privacy, $\delta$ is kept fixed at $10^{-5}$ with $\epsilon$ varied as required and reported. For our model, the privacy budget split is set at 90/10. That is, we reserve 90% of the privacy budget for the differentially private training of the winning ticket and 10% for the differentially private selection of the winning ticket. $\nu$ is kept fixed at 50 for all experiments. Source code for our method is made publicly available[5].

---

[3]https://github.com/google-research/lottery-ticket-hypothesis
[4]https://github.com/tensorflow/privacy
[5]Link available after de-anonymization

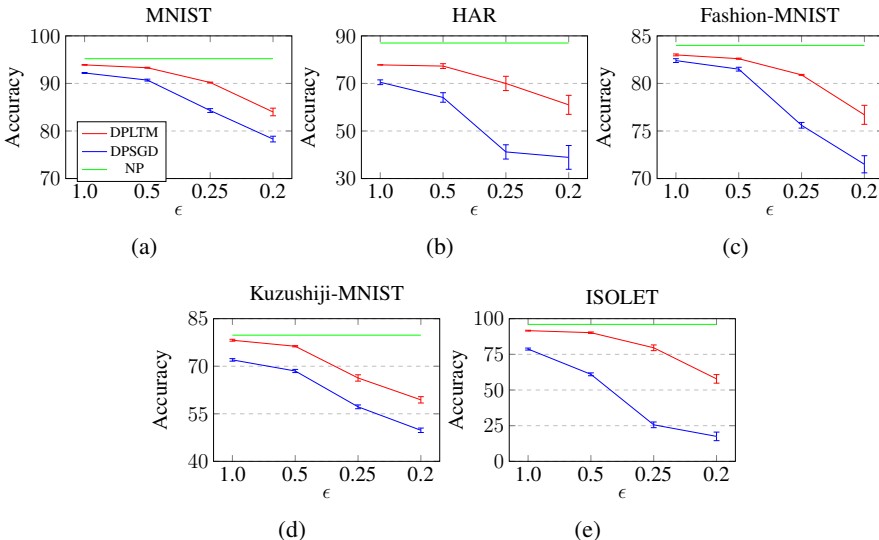

Figure 1: Main comparison: Our proposed method, DPLTM (red line) significantly outperforms our competitor, DPSGD (blue line) on all datasets and for all privacy budgets. The non-noisy model (provided as an upper bound on the performance achievable using this architecture) is presented as the green line. Error bars on the plot represent the standard deviation.

### 4.3 MAIN COMPARISON

Figure 1 shows the results of our main comparison with DPSGD. First and the obvious observation is that our proposed model (DPLTM, red line) significantly outperforms our competitor (DPSGD, blue line). This provides evidence for our earlier claim that our proposed method provides better utility than DPSGD. The second observation is as the privacy budget gets tighter ($\epsilon$ decreases), the performance gap between our proposed method (DPLTM) and DPSGD increases. That is, DPLTM is "robust" compared to DPSGD. The performance gap is specifically larger for small datasets (HAR and ISOLET), as naive DPSGD suffers from worse utility degradation when dataset size is small, due to the interplay between the clipping and sampling probability (See Algorithm 1 and Theorem 1 of Abadi et al. (2016)).

We observe this utility boost due to the reasons discussed in the introduction. That is, in DPLTM, the number of model parameters is "small" compared to the full model used for DPSGD[6]. Hence the "clipping" has a relatively diminished impact on DPLTM's performance compared to DPSGD. Leading to overall better utility and robust models.

### 4.4 CONVERGENCE AND EARLY STOPPING

As we briefly mentioned earlier, DPLTM provides *faster* convergence compared to DPSGD, and hence allows for early stopping with reduced consumption of the total privacy budget. This is made possible by keeping track of the privacy loss at each iteration and stopping when the desired accuracy/privacy budget is reached. Here we investigate the claim in detail. We keep the experimental setup the same as in the previous section, with the privacy budget fixed at $\epsilon = 1, \delta = 10^{-5}$. Figure 2 shows the results. Main plots show the accuracy on the test set as a function of the number of epochs. To focus on the convergence during earlier iterations (when privacy consumption is low), we have provided "inset" plots that are the zoomed version of the highlighted area on the main plot (including first 15 epochs).

We observe that irrespective of the dataset, DPLTM converges faster compared to DPSGD. The highlighted area in the plots shows the scenario if we were to use early stopping, at the point when the total privacy consumption is $\epsilon = 0.3$. In the inset plot, the black dots represent the points where both models (DPLTM and DPSGD) will consume that privacy budget. As DPLTM has extra privacy

---

[6]DPLTM consistently selects winning tickets with total parameters $\leq 10\%$ of the full model.

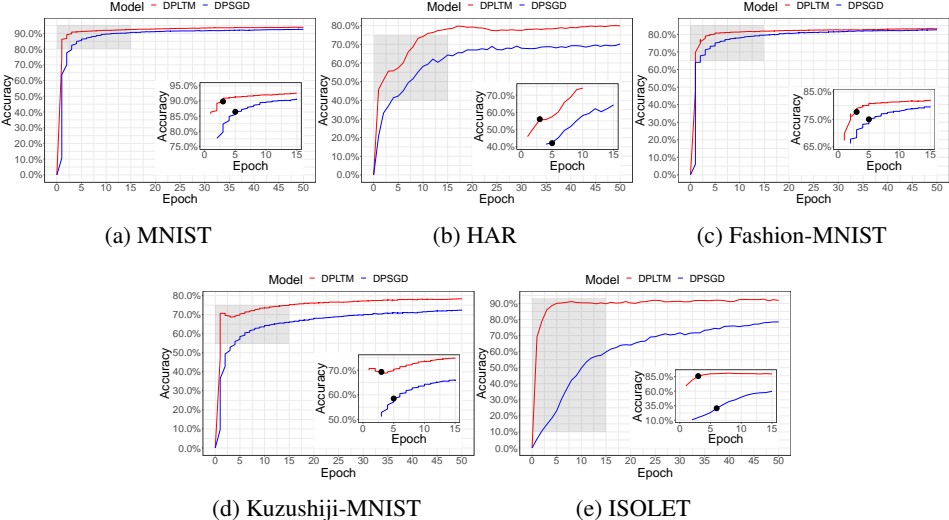

(a) MNIST        (b) HAR        (c) Fashion-MNIST

(d) Kuzushiji-MNIST        (e) ISOLET

Figure 2: Convergence vs the number of epochs for DPLTM and DPSGD. Red line is our proposed method (DPLTM), the blue line is the competitor (DPSGD). The X-axis shows the number of epochs and Y-axis shows the accuracy on the test set. "Inset" plots show the zoomed version of the highlighted area to focus on the plot region when privacy consumption is low. Black dots on the zoomed version signify the points where both models have consumed privacy, $\epsilon = 0.3$. We can observe that DPLTM converges much faster compared to DPSGD with a significant performance difference.

cost of picking the winner, we see that DPLTM will run for "fewer" epochs compared to DPSGD. But, irrespective of the fact that DPLTM runs for fewer epochs, it still outperforms DPSGD by a margin on average of $> 17\%$, which is sometimes greater than $50\%$ (in case of ISOLET). A brief insight into this performance gain is as follows, as the updates in DPLTM are inherently "less noisy" due to reduced number of model parameters, DPLTM converges much faster, using less number of iterations compared to DPSGD.

## 4.5 TICKET TRANSFER

So far, we have seen that our proposed method significantly outperforms DPSGD on all datasets and for all privacy budgets. We use this section to further explore DPLTM. Mainly, we answer the question: As is recently discovered for generic lottery mechanism (Morcos et al. (2019)), that a winning ticket can generalize across datasets, can we extend this notion to DPLTM? Where we can use a publicly available dataset to get a winning ticket in a non-private setting, and then use that winning ticket to train a differentially private model on our sensitive dataset. This has a unique advantage of allowing us to "get rid" of the privacy cost related to the differentially private selection of the winning ticket using EM, and allows us to focus our full privacy budget on the training of the winning ticket. But, sometimes the publicly available datasets are not similar to the sensitive data (different input dimensionality, different domain, different number of outcome classes, etc.). For these scenarios, we investigate the feasibility of using "non-compatible" public datasets for generating lottery tickets. We start with exploring the case where we transfer tickets across the similar architectures (i.e. same dimensionality and outcome classes), and then move on to transfer across non-compatible datasets.

A detailed evaluation is provided in the Appendix. In both cases (Transfer across similar architectures and Transfer across different architectures), DPLTM significantly outperforms DPSGD, especially for tight privacy (small $\epsilon$).

## 5 RELATED WORK

Our related work mainly falls into two categories. First is the prior work related to the lottery ticket mechanism, and second is related to the differential privacy in neural networks. The lottery

ticket hypothesis was introduced in Frankle & Carbin (2018), and provides evidence that there exist subnetworks within a large network, which when trained in isolation, can perform at-par with the large network. This work was further extended in Frankle et al. (2019), where the initial idea was improved to work on larger, deeper networks. Lottery ticket mechanism since has been further explored, it has been shown that the winning tickets can be used across datasets (Morcos et al., 2019), and that the tickets occur in other domains as well, such as in NLP (Yu et al., 2019).

Perturbing the learning process to provide differential privacy has been studied in various contexts (Rajkumar & Agarwal, 2012; Song et al., 2013; Abadi et al., 2016; Shokri & Shmatikov, 2015). Where gradients are perturbed during the gradient descent, so the resulting weight updates, and hence the model itself is differentially private. Differentially private stochastic gradient descent (DPSGD), proposed in Abadi et al. (2016), is the most popular and most often used method for differentially private training for a wide variety of neural networks (Xie et al., 2018; Beaulieu-Jones et al., 2017; McMahan et al., 2017). DPSGD, however, falls short on the utility front, as the noise required for preserving privacy in DPSGD scales up proportional to the model size, discussed at length in the Introduction.

## 6 CONCLUSION

We have proposed DPLTM, an end-to-end differentially private version of the lottery ticket mechanism. Using our custom score function to "pick" differentially private winning tickets, we have shown that DPLTM significantly outperforms DPSGD on a variety of datasets, tasks, and privacy budgets. We have shown that DPLTM converges faster compared to DPSGD, leading to reduced privacy budget consumption with improved utility if early stopping is desired. We have further shown that tickets in DPLTM are transferable across datasets, architectures, and domains. We would like to focus our future work on the detailed study of the mechanism when used for differentially private "transfer-learning" and to further improve the utility guarantees along with the extensions to various "other" models such as the Generative Adversarial Networks (Goodfellow et al., 2014).

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

# A APPENDIX

## A.1 TICKET TRANSFER

### A.1.1 TRANSFER ACROSS SIMILAR ARCHITECTURES

To test the transfer across similar architectures, we use the Kuzushiji-MNIST dataset to select a winning ticket[7], and then use the winning ticket to train a differentially private model for MNIST and Fashion-MNIST. Figure 3 shows the results. We observe that in the beginning, when the privacy is "loose", that is, in case of less noise, DPLTM trained on a ticket from "another" similar dataset performs similar to DPSGD. But, as the privacy gets tight (decreasing $\epsilon$), DPLTM outperforms DPSGD by a significant margin.

---

[7]Winning ticket, in this case, is the smallest model closest to the test accuracy on the full model, such that the model parameters in the small model are $\leq 10\%$ of the full model.

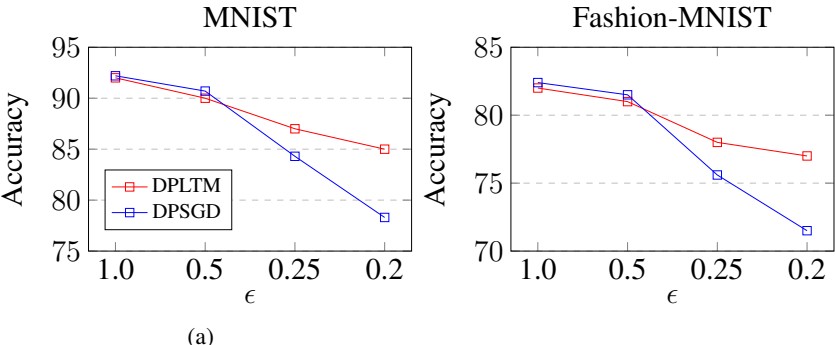

(a)

Figure 3: DPLTM transfer across similar architectures. Red line (DPLTM) is our method, whereas blue line (DPSGD) is the competitor. We can see as the privacy cost increases, our method, trained on a ticket from Kuzushiji-MNIST, outperforms DPSGD by a significant margin.

### A.1.2  TRANSFER ACROSS DIFFERENT ARCHITECTURES

Now we investigate an interesting case, where the publicly available dataset is "quite" different from our sensitive data. Not only in the terms of the domain but also the dimensionality and outcome. To test this scenario, we use the same, Kuzushiji-MNIST dataset to create a winning ticket, and we use the winning ticket to train differentially private models for HAR and ISOLET datasets. As the dimensionality and the output are different between Kuzushiji-MNIST, HAR, and ISOLET, we add an extra layer to the network during ticket generation, which acts as a projection layer to match the dimensionality of HAR or ISOLET. Then during the differentially private training, we remove the top-most layer (projection layer) and the output layer from the selected ticket.

Figure 4 shows the results of this evaluation. We observe a similar trend as before, where we observe similar performance between DPLTM and DPSGD when privacy is "loose", but DPLTM significantly outperforms DPSGD as we decrease $\epsilon$ (tight privacy). The performance, however, is not "as good" as in the case of transfer across similar datasets. Which is intuitive as we loose some "signal" by crossing the domain and dropping layers from the winning ticket. We leave further exploration of this idea for future studies.

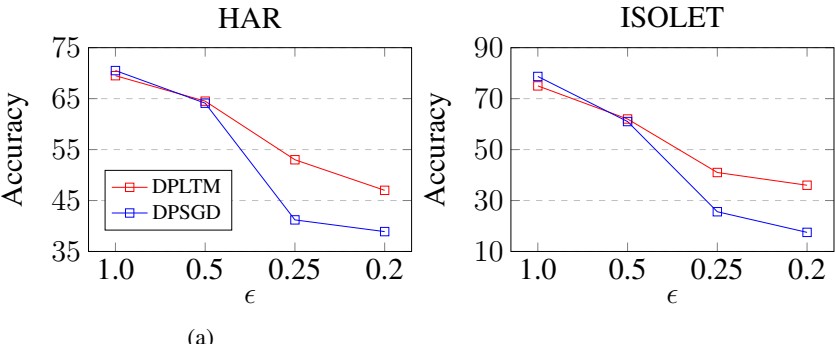

(a)

Figure 4: DPLTM transfer across different datasets and domains. We observe that as the privacy cost increases, our method (DPLTM, red line), significantly outperforms DPSGD (blue line)

### A.2  COMPARISON WITH RANDOM TICKETS

One of the questions a reader might have is "What is the impact of using our score function for selecting a winning ticket?". That is, are we doing much better than using a randomly selected ticket. We use this section to explore this question empirically. We use the tightest reported privacy budget ($\epsilon = 0.2$), and we compare the average accuracy achieved by our proposed method using the "winning tickets" compared to a randomly selected ticket.

Table 2 shows the results. We observe that using the winning ticket via our method has signifi-

| Dataset | Accuracy (Winning Ticket) | Accuracy (Random Ticket) |
|---|---|---|
| MNIST | 0.84 | 0.79 |
| HAR | 0.61 | 0.56 |
| Fashion-MNIST | 0.77 | 0.73 |
| Kuzushiji-MNIST | 0.60 | 0.54 |
| ISOLET | 0.58 | 0.33 |

Table 2: Comparing accuracy of our winning ticket and a randomly sampled ticket. We observe that using the winning ticket has significant advantage over the use of a randomly sampled ticket.

cant advantage over the use of randomly sampled ticket, with our winning ticket outperforming the randomly sampled ticket on every dataset.

### A.3 PROOFS

**Lemma 1.** *Sensitivity ($\Delta$) of the score function, $\mathcal{S}$, is directly proportional to $\nu$.*

*Proof.* We can write the score function as

$$\begin{aligned} \mathcal{S}(\mathcal{C}, \mathcal{A}) &= \mathcal{A} \times (1 - (\nu \mathcal{C})) \\ &= \mathcal{A} - \mathcal{A}\nu\mathcal{C} \end{aligned} \tag{3}$$

Sensitivity is defined as

$$|\max((\mathcal{A} - \mathcal{A}\nu\mathcal{C})) - (\mathcal{A}' - \mathcal{A}'\nu\mathcal{C}'))| \tag{4}$$

where $\mathcal{A}', \mathcal{C}'$ are "neighbouring" to $\mathcal{A}, \mathcal{C}$.

$$\begin{aligned} &|\max(\mathcal{A} - \mathcal{A}\nu\mathcal{C} - \mathcal{A}' + \mathcal{A}'\nu\mathcal{C}')| \\ &\leq |\max(\mathcal{A} - \mathcal{A}' - \mathcal{A}\nu\mathcal{C} + \mathcal{A}'\nu\mathcal{C}')| \\ &\leq |\max(1 - \nu(\mathcal{A}\mathcal{C} - \mathcal{A}'\mathcal{C}'))| \end{aligned} \tag{5}$$

which for the worse case scenario, we get $\mathcal{S}(\mathcal{C}, \mathcal{A}) = |1 - \nu|$ □

**Theorem 1.** *Phase 2 (Selecting a winning ticket) is ($\epsilon_1$) - differentially private.*

*Proof.* We consider the scenario where the EM outputs some element $r \in \mathcal{R}$ on two neighbouring datasets, $X, X'$.

$$
\frac{Pr[\mathcal{M}(X, u, \mathcal{R}) = r]}{Pr[\mathcal{M}(X', u, \mathcal{R}) = r]} = \frac{\left(\dfrac{\exp(\dfrac{\epsilon_1 u(X, r)}{2\Delta u})}{\sum_{r' \in \mathcal{R}} \exp(\dfrac{\epsilon_1 u(X, r')}{2\Delta u})}\right)}{\left(\dfrac{\exp(\dfrac{\epsilon_1 u(X', r)}{2\Delta u})}{\sum_{r' \in \mathcal{R}} \exp(\dfrac{\epsilon_1 u(X', r')}{2\Delta u})}\right)}
$$

$$
= \left(\frac{\exp(\dfrac{\epsilon_1 u(X, r)}{2\Delta u})}{\exp(\dfrac{\epsilon_1 u(X', r)}{2\Delta u})}\right) \cdot \left(\frac{\sum_{r' \in \mathcal{R}} \exp(\dfrac{\epsilon_1 u(X, r')}{2\Delta u})}{\sum_{r' \in \mathcal{R}} \exp(\dfrac{\epsilon_1 u(X', r')}{2\Delta u})}\right)
$$

$$
= \exp\left(\frac{\epsilon_1 (u(X, r') - u(X', r'))}{2\Delta u}\right) \cdot \left(\frac{\sum_{r' \in \mathcal{R}} \exp(\dfrac{\epsilon_1 u(X, r')}{2\Delta u})}{\sum_{r' \in \mathcal{R}} \exp(\dfrac{\epsilon_1 u(X', r')}{2\Delta u})}\right)
$$

$$
\leq \exp(\frac{\epsilon_1}{2}) \cdot \exp(\frac{\epsilon_1}{2}) \cdot \left(\frac{\sum_{r' \in \mathcal{R}} \exp(\dfrac{\epsilon_1 u(X, r')}{2\Delta u})}{\sum_{r' \in \mathcal{R}} \exp(\dfrac{\epsilon_1 u(X', r')}{2\Delta u})}\right)
$$

$$
\leq \exp(\epsilon_1)
$$

$\square$

