# OpenReview forum: "Winning Privately: The Differentially Private Lottery Ticket Mechanism"
_ICLR.cc/2020/Conference — Reject_

### Official Review · AnonReviewer1 · 2019-10-22
**Official Blind Review #1**

**Rating:** 3

**Review:**

This paper is a mash-up of recent work on differentially private stochastic gradient descent (DPSGD) and the lottery ticket hypothesis. Differential privacy is a paradigm for ensuring that statistical models learned from a large dataset do not disclose any particulars of individual elements of that dataset. DPSGD is a technique that applies differential privacy ideas to models trained with stochastic gradient descent, where privacy is guaranteed by clipping and adding noise to gradients. On the other hand, the lottery ticket hypothesis is a method for finding sparse sub-networks contained in larger dense neural network models that are at least as accurate as the underlying dense model. The authors propose and demonstrate that by combining DPSGD with the lottery ticket hypothesis that they can train end-to-end differentially private models that outperform the DPSGD technique.

I am up in the air about whether or not to accept this paper. I have trouble assessing the originality and significance of this work, the former since it seems to be a simple combination of two recent ideas, and the latter because I don't really have a context for how important the development of differentially-private optimization strategies are. I also didn't really find the experimental results to be that extensive, but I don't have a real sense of what state of the art is for differential privacy. Perhaps some of these issues could be mitigated by some additional exposition? I think one of my problems is that most of the paper is written to show that this new technique is superior to DPSGD, but I don't really have a good sense if either of them are sufficient as a practical solution of this problem.

Some additional questions I had are as follows:

1) One of the reasons given for combining the lottery ticket hypothesis with DPSGD is that DPSGD scales poorly as models get larger due to clipping by the norm of the gradient. Could you not compensate for this by scaling the norm by the number of parameters, or does that mess up the differential-privacy calculation?

2) I don't understand how the winning ticket selection portion is differentially privacy. I understand that they use the exponential mechanism, but I would have assumed that the dataset needs to be varied in addition to the model. Is there an intuitive explanation for how privacy is achieved essentially just by varying the initialization and not actually adding noise to the data?

3) Are there any datasets or methods where the effectiveness of the differential privacy technique could be directly assessed? For example, is there a way of testing whether the model is memorizing specific data elements, and then further showing that these differential privacy techniques mitigate this? This would be a nice check to have in addition to the plots of accuracy vs. privacy budget.

**Experience Assessment:**

I do not know much about this area.

**Review Assessment: Checking Correctness Of Derivations And Theory:**

I assessed the sensibility of the derivations and theory.

**Review Assessment: Checking Correctness Of Experiments:**

I assessed the sensibility of the experiments.

**Review Assessment: Thoroughness In Paper Reading:**

I read the paper at least twice and used my best judgement in assessing the paper.

---

> ### Author Response · Authors · 2019-11-08
> **Author response to R-1**
>
> We thank the reviewer for their time and we sincerely hope we can clarify all doubts and provide clear answers to all questions.
>
> To start, we would like to emphasize the significance of our work, our work is more than a mere mash-up of recent works. Our contribution lies in developing an end-to-end differentially private training paradigm for neural networks, which significantly improves upon the current state-of-the-art, both in terms of utility and privacy, using a fraction of the parameters of the full model. One of the main goals and a core research area in differentially private machine learning is to investigate and improve the privacy-utility trade-off. Our method, being accessible and providing significantly better utility with tight privacy guarantees aims to boost the real-world applicability of differentially private machine learning methods, which to date has enjoyed limited success, mainly due to the problems with limited utility.
>
> Next, we provide answers to the specific questions raised by the respected reviewer.
>
> Q1) One of the reasons given for combining the lottery ticket hypothesis with DPSGD is that DPSGD scales poorly as models get larger due to clipping by the norm of the gradient. Could you not compensate for this by scaling the norm by the number of parameters, or does that mess up the differential-privacy calculation?
>
> A1) Differential privacy guarantees in DPSGD [1] are based on bounding the impact of per-observation gradient via gradient clipping using the l2 norm, scaled by a constant $C$, which then has a multiplicative effect on total noise ($C \sigma$). Using a larger scaling parameter will directly lead to larger noise, thus diminishing utility.
>
>
> Q2) I don't understand how the winning ticket selection portion is differentially privacy. I understand that they use the exponential mechanism, but I would have assumed that the dataset needs to be varied in addition to the model. Is there an intuitive explanation for how privacy is achieved essentially just by varying the initialization and not actually adding noise to the data?
>
> A2) We would like to emphasize that this step is just selecting the winning architecture, that is then trained with differential privacy from scratch. To ensure that the selection of the architecture is done in a differentially private fashion, we use the exponential mechanism with our custom score function. The exponential mechanism selects the winner with probability defined by the score function, in which the design of our custom score function ensures that we select "good" candidates with high probability. Simply put, the exponential mechanism selects the winners with "noisy" probability, not a "noisy winner" [2]. This is the part of the appeal of using the exponential mechanism, leading to its applicability in problems such as selecting max, top-k items, etc. with differential privacy.
>
>
> Q3) Are there any datasets or methods where the effectiveness of the differential privacy technique could be directly assessed? For example, is there a way of testing whether the model is memorizing specific data elements, and then further showing that these differential privacy techniques mitigate this? This would be a nice check to have in addition to the plots of accuracy vs. privacy budget.
>
> A3) There are some proposed methods such as the membership inference attack [3], that have been studied with DPSGD [4] and that have shown that DPSGD does indeed offer privacy protection to the contributing users in the training dataset, when the privacy guarantees are tight (small $\epsilon$), but this also leads to the utility degradation, result of the privacy-utility trade-off, which our proposed method significantly improves upon. As our final winning ticket trains using the DPSGD method, we do not find it is necessary to evaluate the models against such attacks, especially when the results are already readily available. Instead, we focus on investigating and providing details on the significantly improved utility compared to the naive DPSGD.
>
> [1] Abadi, Martin, et al. "Deep learning with differential privacy." Proceedings of the 2016 ACM SIGSAC Conference on Computer and Communications Security. ACM, 2016.
> [2] Dwork, Cynthia, and Aaron Roth. "The algorithmic foundations of differential privacy." Foundations and Trends® in Theoretical Computer Science 9.3–4 (2014): 211-407.
> [3] Shokri, Reza, et al. "Membership inference attacks against machine learning models." 2017 IEEE Symposium on Security and Privacy (SP). IEEE, 2017.
> [4] Rahman, Md Atiqur, et al. "Membership Inference Attack against Differentially Private Deep Learning Model." Transactions on Data Privacy 11.1 (2018): 61-79.

---

### Official Review · AnonReviewer2 · 2019-10-23
**Official Blind Review #2**

**Rating:** 3

**Review:**

The existing differentially private training mechanism of deep neural networks includes a step that applies L2 normalization to gradient vectors. The authors argued that such norm clipping could lead to a small norm model parameter at the end, and the noise can thus overwhelm the accumulated gradient, leading to a substantial loss in terms of utility. To mitigate such utility loss, the authors proposed to use the lottery ticket mechanism to train a smaller network with similar utility and design an end-to-end differentially private mechanism for the whole process, including the selection of sub-network and the re-training of the selected sub-network. The general idea is quite interesting, but I have the following concerns.

-   From Algorithm 1 of Abadi et al. 2016, one could see that the norm of per instance gradient is at least C, and the added noise has standard deviation of \sigma C. Furthermore, since the final gradient step is an accumulation of all the per instance gradient in the mini-batch, this means that the scale of the final gradient is roughly $B\cdot C$, where $B$ is the batch size. Note that since the noise is added to the accumulation gradient, so the scale of the noise should be roughly 1/B of the accumulated gradient. It seems to me that in this case the injected noise shouldn't overwhelm the gradient vector. Could the authors have a comment on this?

-   The overall algorithm can be understood as a simple composition of the Exponential Mechanism algorithm and the DPSGD algorithm. Both of them satisfy differential privacy, so by a simple composition theorem argument the whole process also satisfies DP. This all makes sense, but the novelty is quite limited. The authors claim that the proposed mechanism is
"different from DPSGD's naive implementation". This is not true, since at the end the original DPSGD algorithm is still used, just on a subnetwork.

-   If the goal is to train a model with fewer parameters and comparable utility, why not simply starting from a small model with DPSGD and then use the large model as a teaching model? This simple strategy naturally gives DP guarantee without worrying about the LTH, hence you can also get same privacy guarantee with less injected noise.

-   The experimental results basically confirm the expectation that with less injected noise, the model achieves better utility as compared with the DPSGD applied on the original large model.

**Experience Assessment:**

I have read many papers in this area.

**Review Assessment: Checking Correctness Of Derivations And Theory:**

I assessed the sensibility of the derivations and theory.

**Review Assessment: Checking Correctness Of Experiments:**

I assessed the sensibility of the experiments.

**Review Assessment: Thoroughness In Paper Reading:**

I read the paper at least twice and used my best judgement in assessing the paper.

---

> ### Author Response · Authors · 2019-11-08
> **Author response to R-2**
>
> We thank the reviewer for their time and comments, and for appreciating the importance of our work. We provide our answers next.
>
> Q1)  From Algorithm 1 of Abadi et al. 2016, one could see that the norm of per instance gradient is at least C, and the added noise has standard deviation of \sigma C....
>
> A1) Yes, in Abadi et al.’s [1] Algorithm 1, we know that the norm of the per-observation gradient is bounded by $C$, which enforces the sensitivity required for differential privacy. However, as the reviewer has mentioned, the noise reduction (in Algorithm 1 of [1]) depends on the group size (minibatch) (i.e. large group size provides a “noise reduction” effect), but at the same time, large group size also directly adversely impacts the noise scale ($\sigma$), see Theorem 1 of [1], where $q$ has a scaling effect on $\sigma$, where we get a smaller $q$ with smaller group size and vice-versa, this makes the naive DPSGD perform particularly worse on “smaller” datasets, see the performance gap for dataset ISOLET and HAR as an example (Figure 1 of our paper).
>
> This phenomenon creates an interesting dilemma, where increasing batch size provides a "noise reduction" effect due to summation in Algorithm 1 [1], but at the same time increases the noise scale (Theorem 1 [1]). Abadi et al. in the paper recommend using a smaller batch size for the same reason. In our case, as we are working with a very small fraction of model parameters compared to the full model, our method enjoys significantly better utility while simultaneously providing tight privacy. We have added this detail to our main comparison (Section 4.3).
>
>
> Q2) The overall algorithm can be understood as a simple composition of the Exponential Mechanism algorithm and the DPSGD algorithm.....
>
> A2) We consider DPSGD as a generic training algorithm, which has been used widely across models (such as differentially private GANs [2][3], adversarial examples [4], various versions of differentially private neural networks [5], etc.). Our contribution is not based on the argument that we propose a radical new paradigm, but it is that we show how to significantly improve the existing method, which has direct implications on a wide array of methods using the naive method. We endeavored to make the algorithm and the paper as clear and as simple as possible, to ensure that it is easily accessible and applicable by a wide variety of practitioners, with a goal to boost the privacy-preserving machine learning in practice, prior successes of which have been limited, mainly due to the privacy-utility trade-offs of the current methods (which we propose to significantly improve upon).
>
>
>
> Q3)  If the goal is to train a model with fewer parameters and comparable utility, why not simply starting from a small model with DPSGD and then use the large model as a teaching model? ....
>
> A3) There are several issues with this approach. First, starting from a random small model does not guarantee that the small model will provide good utility. This was the driving force behind the lottery ticket hypothesis (to find small, high utility models that can be trained from scratch in isolation). Second, when we move to the student-teacher paradigm, it is a different research direction, not our focus in this work, we propose methods to significantly improve the performance of DPSGD, without constraining the training setup, so that our approach can be used anywhere, where DPSGD is applicable.
>
>  Further, the student-teacher methods explored in previous models such as PATE [6] have their limitations. Such as the utility in PATE depends on the number of disjoint partitions, and the availability of public data to start the training process. This does not work in settings where we only have private sensitive data (in most scenarios requiring privacy protection), and in scenarios where the datasets are small (creating many disjoint partitions is not feasible). Our proposed method does not have any such limitations.
>
>
> [1] Abadi, Martin, et al. "Deep learning with differential privacy." Proceedings of the 2016 ACM SIGSAC Conference on Computer and Communications Security. ACM, 2016.
> [2] Esteban, Cristóbal, Stephanie L. Hyland, and Gunnar Rätsch. "Real-valued (medical) time series generation with recurrent conditional gans." arXiv preprint arXiv:1706.02633 (2017).
> [3] Beaulieu-Jones, Brett K., et al. "Privacy-preserving generative deep neural networks support clinical data sharing." Circulation: Cardiovascular Quality and Outcomes 12.7 (2019): e005122.
> [4] Lecuyer, Mathias, et al. "Certified robustness to adversarial examples with differential privacy." 2019 IEEE Symposium on Security and Privacy (SP). IEEE, 2019.
> [5] McMahan, H. Brendan, et al. "Learning differentially private recurrent language models." arXiv preprint arXiv:1710.06963 (2017).
> [6] Papernot, Nicolas, et al. "Scalable private learning with pate." arXiv preprint arXiv:1802.08908 (2018).

---

### Official Review · AnonReviewer3 · 2019-10-23
**Official Blind Review #3**

**Rating:** 3

**Review:**

This paper proposes a differentially private version of the lottery ticket mechanism using the exponential mechanism, thus improving the utility of DPSGD by reducing the number of parameters. It provides the privacy guarantee of the proposed algorithm and shows experimentally that the proposed algorithm outperforms DPSGD across datasets and privacy parameters.

The proposed algorithm seems quite interesting. Though it is more of a simple combination of the non-private lottery ticket mechanism and the exponential mechanism, improving utility for DPSGD is a very important topic in differentially private machine learning. The experimental results seem pretty strong.

My only concern is on the aspect of privacy, specially Lemma 1. I think if you’re only using the fact that A and C are in [0, 1], then A*(1-nu*C) can change by 1 if you go from (A=0, C=0) to (A=1, C=0). In the last step of equation (5), you replaced A-A’ by 1, which I think needs to be double-checked since there is the absolute value outside (and I guess the equality should  be <=).
If the calculation is correct, I’m still a bit concerned that the sensitivity is a bit too high compared to the signal. A and C are in [0, 1], and it seems like the sensitivity may not be much smaller than 1, which means the exponential mechanism can be pretty random. To that end, I think you may consider experimentally comparing with DPSGD with a randomly selected ticket, or a ticket with a moderate number of parameters kept.


----- Post-rebuttal response -----
I still don't see why the sensitivity is |1-\nu|. (Doesn't that mean sensitivity is 0 when \nu=1?) If we have (A=1, C=0) and (A'=0, C'=0), we have S(A,C)-S(A',C')=1-0=1; if we have (A=1, C=0) and (A'=1, C'=1), we have S(A,C)-S(A',C')=1-(1-\nu)=\nu. Yet for \nu<1, |1-\nu|<1 and for \nu>1, |\nu-1|<\nu. I believe the sensitivity should be something like max(\nu, 1). You may want to further check that. And I still think this is because the last step in (5) is not correct. Since you have absolute sign, you can't do |A-A'-X| \leq ||A-A'|-X|.

I'm also confused by the range of \nu I should think of.
If \nu <= 1, then in the example you give (A = 0.8, C=0.9,0.7,0.4,0.1, eps=0.1), I think the probability distribution used by exponential mechanism is not much higher for 0.9 than 0.1 (I think it's roughly 0.22 vs. 0.29?) with sensitivity |1-\nu|.
If \nu > 1, then I don't quite understand the score function S(A,C) = A*(1-\nu*C), since for C > 1/\nu, (1-\nu*C) is negative and S would thus penalize higher A. The higher \nu is, the wider the range of C that S will penalize. I can sort of understand this as setting a threshold such that any C above that threshold is just considered useless, but I still think it's a bit unreasonable to penalize higher accuracy, even for very complex model. You may want to explain more on that end.

Thanks for the experiments! They're very useful in supporting your proposed algorithm.


**Experience Assessment:**

I have read many papers in this area.

**Review Assessment: Checking Correctness Of Derivations And Theory:**

I assessed the sensibility of the derivations and theory.

**Review Assessment: Checking Correctness Of Experiments:**

I assessed the sensibility of the experiments.

**Review Assessment: Thoroughness In Paper Reading:**

I read the paper at least twice and used my best judgement in assessing the paper.

---

> ### Author Response · Authors · 2019-11-08
> **Author response to R-3**
>
> We thank the reviewer for their time and for appreciating the simplicity and significance of our proposed method. Next, we provide answers to the specific questions raised by the reviewer.
>
> Q) My only concern is on the aspect of privacy, specially Lemma 1. I think if you’re only using the fact that A and C are in [0, 1], then A*(1-nu*C) can change by 1 if you go from (A=0, C=0) to (A=1, C=0). In the last step of equation (5), you replaced A-A’ by 1, which I think needs to be double-checked since there is the absolute value outside (and I guess the equality should be <=). If the calculation is correct, I’m still a bit concerned that the sensitivity is a bit too high compared to the signal. A and C are in [0, 1], and it seems like the sensitivity may not be much smaller than 1, which means the exponential mechanism can be pretty random. To that end, I think you may consider experimentally comparing with DPSGD with a randomly selected ticket, or a ticket with a moderate number of parameters kept.
>
> A) Yes, you are correct, the equality should be $\le$, we apologize for the oversight and we have changed it in the revised version. Regarding the $A-A’$, we claim that the final sensitivity is "proportional" to $\nu$, where $\nu$ is a much larger number compared to $A$ ($\nu \gg A$), hence any change in $A-A’$ does not impact our final results.
>
> As for the sensitivity and its impact on the utility. We have designed the score function in such a way that our outcomes of interest (models with a small number of parameters and high utility) have a high probability of selection (we discuss this briefly in Section 3.4). Consider this simple example: Let's say we have four models, all with the same utility (accuracy score of 0.8), but a varying number of model parameters (proportion of retained parameters is 0.9,0.7,0.4,0.1). Using our custom score function with $\epsilon = 0.1$, we can observe that the probability of selection for the model with 0.1 proportion of the original model’s parameters is the highest. That is, our custom score function ensures that the Exponential Mechanism in our case does not degenerate to uniform random sampling, we have added this detail in Section 3.4 of our revision.
>
> The smallest ticket with the highest accuracy provides the best "end-utility" for our method, as we have discussed at length in our paper. With the design of the score function specifically geared towards the selection of such tickets with high probability, randomly sampled tickets in expectation will not provide good utility compared to the winners selected by our method. However, to support our claim, we have added extra results. Please see the table below for such comparison, where we compare the two methods (selecting the best ticket with our method and a randomly selected ticket) at the tightest reported privacy budget ($\epsilon=0.2$), showing the positive impact of winners selected via our proposed method.
>
> ——————————————————————--
> Dataset                 DPLTM        Random Ticket
> ——————————————————————--
> MNIST                    0.84             0.79
> HAR                        0.61             0.56
> Fashion-MNIST    0.77              0.73
> Kuzushiji-MNIST  0.60              0.54
> ISOLET                   0.58              0.33
>
> We have added the results in the appendix of the revised paper (Section A.2).

---

### Author Response · Authors · 2019-10-17
**Typo**

There is a typo in Algorithm 1, phase 2, line 17; and Section 3.3.1, Eqn. 2. The parentheses after $\exp$ should enclose the $2 \Delta$ term, i.e. $P = \frac{\exp(\frac{\epsilon_1 \mathcal{S}}{2 \Delta})}{\sum_{T} \exp(\frac{\epsilon_1 \mathcal{S}}{2 \Delta})}$. We apologize for the inconvenience.

---

### Decision · Program_Chairs · 2019-12-19

**Decision:**

Reject

**Comment:**

This paper provides an approach to improve the differentially private SGD method by leveraging a differentially private version of the lottery mechanism, which reduces the number of parameters in the gradient update (and the dimension of the noise vectors). While this combination appears to be interesting, there is a non-trivial technical issue raised by Reviewer 3 on the sensitivity analysis in the paper. (R3 brought up this issue even after the rebuttal.) This issue needs to be resolved or clarified for the paper to be published.